

# Physical activity and self-efficacy in college students: the mediating role of grit and the moderating role of gender

Hongyan Yu[1,*], Tingfei Zhu[2,*], Jianing Tian[1], Gang Zhang[3], Peng Wang[1], Junxiong Chen[1] and Liqun Shen[1]

[1] Department of Physical Education, Shanghai Jiao Tong University, Shanghai, China
[2] Psychological Counseling Center, Shanghai Jiao Tong University, Shanghai, China
[3] Shenzhen Shangbu Middle School, Shenzhen, China
[*] These authors contributed equally to this work.

## ABSTRACT

**Background**. There is a paucity of knowledge concerning the psychological variables that serve to facilitate the connection between physical activity and self-efficacy, and the factors capable of moderating these pathways. This study aimed to examine the relationship between physical activity and self-efficacy among college students, with a focus on the mediating effect of grit and the moderating effect of gender.

**Methods**. This study recruited 3,228 undergraduate students from a university in Shanghai, China. They completed the General Self-Efficacy Scale, the Short Grit Scale, and the International Physical Activity Questionnaire. Statistical analysis was conducted using SPSS 26.0 and the Process v4.0 plugin.

**Results**. Physical activity had both a direct effect on self-efficacy ($\beta = 0.07$, 95% CI [0.04–0.11]) and an indirect effect through the two dimensions of grit: perseverance of effort ($\beta = 0.06$, 95% CI [0.04–0.07]) and consistency of interest ($\beta = 0.03$, 95% CI [0.02–0.04]). The mediating effect explained 53.27% of the total effect. Furthermore, gender moderated the relationship between perseverance of effort and self-efficacy, with a stronger effect observed in males ($\beta = 0.08$, $t = 3.27$, $p < 0.01$).

**Conclusion**. The results revealed that grit is an underlying psychological mechanism that links physical activity and self-efficacy. Moreover, gender moderates the effect of perseverance of effort on self-efficacy, with a stronger effect observed in males. These findings have practical implications for educators to design tailored physical activity interventions that foster grit and self-efficacy among college students.

## INTRODUCTION

Self-efficacy is the core concept of social cognitive theory, which pertains to an individual's belief in their capacity to attain anticipated goals and accomplishments. This belief serves as motivation for behavior, inspiring individuals to enhance their abilities to achieve success (*Bandura, 1986*). *Bandura (1978)* emphasizes that self-efficacy not only affects individuals' belief in their ability to cope with stress but also directly impacts their psychosomatic regulation system, thereby affecting their physical and mental health. A

Corresponding author
Liqun Shen,
shenliqun911@sjtu.edu.cn

meta-analysis involving 242,023 students confirmed the significant positive impact of self-efficacy on achievement competencies, skills, knowledge, and health behaviors (*Çikrıkci, 2017*). The study by *Gore Jr (2006)* suggests that self-efficacy is an important indicator for predicting the success of adolescents. Furthermore, in the context of the COVID-19 pandemic, the Protection Motivation Theory suggests that self-efficacy is the most influential determinant in an individual's transition from recognizing a health threat to adopting a health behavior (*Plotnikoff et al., 2010*). Therefore, investigating methods to enhance individuals' self-efficacy is crucial.

Physical activity (PA) could serve as a critical factor influencing individual self-efficacy (*Bandura, 1978*). Physical activity is defined by the World Health Organization (WHO) as any bodily movement produced by skeletal muscles that results in energy expenditure (*World Health Organization, 2010*). Physical activity encompasses a broad spectrum, including structured exercises and everyday activities such as walking, working, and household chores (*Caspersen, Powell & Christenson, 1985*). The self-efficacy theory suggests that an individual's level of Physical activity can influence the development of self-efficacy (*McAuley & Blissmer, 2000*). A strong positive correlation between PA and self-efficacy has been found in the UK (*Darker et al., 2010*), Italy (*Latino et al., 2021*), and the USA (*Wiedenman et al., 2023*). For example, a six-week walking intervention led to a significant increase in self-efficacy among adult participants (*Darker et al., 2010*). A 12-week study of moderate to vigorous aerobic exercise had a positive and significant impact on the self-efficacy of overweight female students (*Latino et al., 2021*; *Wu, Zhao & Zhao, 2022*). Similar findings were found in the Chinese university student population (*Wu, Zhao & Zhao, 2022*; *Han et al., 2022*). One review study concluded that self-efficacy is an outcome of PA (*McAuley & Blissmer, 2000*). Thus, a promising avenue for college students to enhance self-efficacy is to begin with PA.

Although PA possesses the potential to enhance self-efficacy, recent studies have revealed that this strengthening process may not be consistently stable (*Bandura, Freeman & Lightsey, 1999*). *Dinger & Behrens (2006)* found that wearing the ACTiGraph wGT3X-BT accelerometer for seven consecutive days revealed differences in the PA levels of Portuguese university students between weekdays and weekends. Otherwise, self-efficacy is a dynamic belief system that is shaped by specific situations and may be influenced by subjective experiences related to health (*Bandura, Freeman & Lightsey, 1999*; *Latino et al., 2023*). *Blanchard et al. (2007)* reported that the positive relationship between PA and task self-efficacy may become increasingly unstable over time. Therefore, there is a need to find a mechanism that can stabilize PA to promote self-efficacy. Multiple hypotheses from diverse disciplinary perspectives, including the distractor theory, self-efficacy theory, social interaction hypothesis, endorphin hypothesis, and monoamine hypothesis, have been put forward to elucidate how PA boosts self-efficacy (*Paluska & Schwenk, 2000*). However, the precise mechanism underlying this relationship remains elusive, emphasizing the theoretical nature of this association and the necessity for continued exploration. In empirical research, a number of scholars have introduced various mediating variables to elucidate the potential link between PA and self-efficacy. These variables include peer attachment (*Li et al., 2023*), motivation (*Zhu, 2021*), self-esteem (*Kong et al., 2020*),

strength quality (*Yang & Liu, 2013*), coordination (*Wang & Han, 2012*), endurance (*Yang, 2013*), planning (*Zhou et al., 2016*), self-management strategies (*Dishman et al., 2005*), and goal ratings (*Hall et al., 2010*). Nevertheless, none of these variables has demonstrated the ability to consistently facilitate the translation of PA into enhanced self-efficacy.

In prior investigations, grit has been acknowledged as a trait-level conceptualization, marked by a relatively enduring structure that evolves over time (*Hodge, Wright & Bennett, 2018*). Grit is an term used to describe a personality trait that enables individuals to exhibit persistence, stability, and consistency in their behavioral tendencies (*Newland, Gitelson & Legg, 2020*). It is defined as "perseverance and passion in the face of obstacles and adversity in the pursuit of long-term goals," comprising two components: perseverance of effort and consistency of interest (*Duckworth et al., 2007*). Perseverance of effort refers to the trait that enables individuals to persist in the face of setbacks until the goal is achieved, while consistency of interest refers to pursuing the same goals over an extended period (*Duckworth et al., 2007*). It is worth noting the key variable of grit in positive psychology. The research indicates that perseverance of effort and consistency of interest are essential components of motivation with direct implications for behavioral outcomes (*Hagger & Chatzisarantis, 2016*). Strayhorn argues that grit emerges as a more dependable predictor of academic performance compared to intelligence (*Strayhorn, 2014*). Moreover, grit not only predicts greater pandemic resilience and slightly less psychological impact during the pandemic (*Bono, Reil & Hescox, 2020*), it also contributes to maintaining students' mental health amidst these challenging circumstances (*Liu, Ye & Hu, 2022*; *Casali, Feraco & Meneghetti, 2023*).

PA exhibits a robust correlation with grit, as evidenced by insights from brain neuroscience. The dorsomedial prefrontal cortex (dmPFC) emerges as a pivotal brain region linked to grit (*Myers et al., 2016*; *Wang et al., 2017*). One study has shown that acute moderate exercise effectively modulates dmPFC activity (*Yanagisawa et al., 2010*). Furthermore, even moderate-intensity exercise coupled with cognitive demands has the capacity to significantly enhance PFC activity (*Kimura et al., 2022*). Concurrently, exercise adherence relies on grit, as individuals adhering to a regular exercise routine are inclined to possess higher levels of grit (*Reed, Pritschet & Cutton, 2013*). Additionally, heightened PFC activity has also been observed in walking (*Mirelman et al., 2014*). Notably, both low intensity and vigorous PA positively associated with grit (*Dunston et al., 2022*; *Daniels et al., 2023*). In summary, these neuroscientific findings substantiate empirical evidence, underscoring the significant role of PA in fostering grit across diverse intensities and exercise modalities.

Additionally, multiple studies have demonstrated that grit positively impacts self-efficacy across various populations, including elementary school students, pre-service teachers, college student-athletes, and physical education college students (*Lim, Ha & Hwang, 2016*; *Riddle, 2018*; *Kim, 2019*; *Jang & Huh, 2019*). The millennia-old Confucian culture in China profoundly molds the moral character and cultivation of its people, emphasizing values of diligence, perseverance, and dedication in education (*Zhang, 2000*). Within this cultural framework, individuals often bolster their self-efficacy by cultivating tenacious qualities, with grit serving as a pivotal factor in achieving this objective. Furthermore, the concept

of Optimal Performance and Health (OPAH) elucidates how grit enhances achievement and well-being, both integral components of self-efficacy (*Datu, 2021*). Consequently, our study postulates that grit may positively influence self-efficacy among Chinese college students.

In summary, elevating individual's PA level significantly enhances their mental resilience, self-discipline, and capacity to overcome challenges, thereby fostering the development of grit (*Warburton, Nicol & Bredin, 2006*; *Seçer & Yildizhan, 2020*). The grit nurtured through PA provides robust psychological support for individuals confronting various challenges in life, empowering their self-efficacy.

Gender has the potential to moderate the relationship between PA, self-efficacy, and grit. There is contradictory evidence regarding the presence of gender differences in grit (*Hodge, Wright & Bennett, 2018*). Meanwhile, a study has highlighted the issue of cultural bias in assessing grit, which remains unaddressed (*Datu, 2021*). This serves as a reminder to consider the impact of gender issues on relationships involving the grit variable. In traditional Chinese beliefs, the positioning of gender roles has profound effects on both men and women, primarily manifested in societal and cultural shaping as well as expectations. Men are typically expected to demonstrate traits such as competitiveness and resilience, while women are more often expected to exhibit qualities like gentleness and submissiveness (*Wang, Qi & Zhang, 2011*). Additionally, physiological differences also impact gender performance in physical activities, such as variances in muscle mass and skeletal structure (*Miller et al., 1993*). These physiological disparities between genders may also influence individuals' grit and self-efficacy, as individuals often assess their abilities and potential in physical activities based on their own physiological conditions. Tao's study explores gender disparities in the correlation between PA and grit among college students, highlighting a more pronounced influence for females in this pathway (*Tao et al., 2022*). Spence's investigation indicates gender disparities in the relationship between PA and self-efficacy, with females exhibiting a stronger correlation between PA and self-efficacy (*Spence et al., 2010*). However, it remains unclear whether there is a moderating effect of gender on the relationship between grit and self-efficacy (*Webb-Williams, 2014*; *Sigmundsson, Haga & Hermundsdottir, 2020*). Further investigation is needed to explore gender differences in the relationship between grit and self-efficacy. Considering the combined influence of societal and physiological differences, the model proposed in this study regarding PA, grit, and self-efficacy may be subject to gender variations. Therefore, it is imperative to fully consider gender factors to more accurately comprehend individuals' psychological and behavioral traits.

## The present study

The theories of positive psychology and social cognitive theory suggest that the trait of grit plays a significant role on the relationship between physical activity and self-efficacy. Some scholars have also found connections between PA and grit, as well as between grit and self-efficacy. However, the specific mediating role of grit, particularly concerning the consistency of interest and perseverance of effort facets, has not been fully explored. Meanwhile, given the fact that Chinese college students' self-efficacy levels were severely

compromised during the COVID-19 pandemic (*Wang, Teng & Liu, 2023*; *Dai et al., 2022*), our study aims to propose a model that explores the mediating role of grit and the moderating role of gender in the relationship between PA and self-efficacy among Chinese college students. This study is expected to provide theoretical insights and practical strategies to improve personal self-efficacy among college students globally after being affected by COVID-19. Based on the above summary of literature and theoretical analyses, our study proposes the following hypotheses:

Hypothesis 1: Physical activity positively predicts self-efficacy.

Hypothesis 2: Perseverance of effort and consistency of interest act as parallel mediating roles between physical activity and self-efficacy.

Hypothesis 3: Gender moderates the pathway from perseverance of effort and consistency of interest to self-efficacy in the model.

# MATERIALS & METHODS

## Participants

This study focused on undergraduate students from Shanghai Jiao Tong University in China, with a total undergraduate population of 17,460 (*Shanghai Jiao Tong University, 2021*). To ensure the accuracy and representativeness of the research, we employed the RAOSOFT online calculator to determine the minimum required sample size, utilizing a margin of error of 3%, a confidence level of 95%, and a response distribution of 50% (*Raosoft Inc., 2004*). The calculated minimum sample size required was determined to be no less than 1006 individuals. Employing a convenient sampling method, we successfully recruited 3890 students from Shanghai Jiao Tong University, representing 22.28% of the total undergraduate population. The survey subjects of this study are from Shanghai Jiao Tong University, one of the top nine universities in China, with students coming from all over the country, ensuring sample diversity. Additionally, this study employs two aspects for data quality control to minimize sample bias that may result from non-random sampling. Firstly, by increasing the sample size to improve the reliability of statistical results; Secondly, during the survey process, we balanced the distribution of survey subjects in terms of gender, grade, age, etc. Following data collection, we conducted the following data processing steps: (1) removal of samples with missing values in responses; (2) delete responses with homogeneity and regularity, such as all responses to the questions being nearly identical, or exhibiting simple increasing or decreasing sequences; (3) exclusion of samples with a cumulative daily activity time exceeding 960 min, in accordance with the criteria of the International Physical Activity Questionnaire (*Fan, Lyu & He, 2014*). We retained 3228 valid samples, accounting for 82.98% of the contacted students. The age of the participants ranged from 17 years to 28 years old ($M = 19.62$, $SD = 0.84$). There were 2,084 male students (64.6%) and 1,144 female students (35.4%). Of these, 1159 were freshmen (35.9%), 878 were sophomores (27.2%), 630 were juniors (19.5%), and 561 were seniors (17.4%). Signed informed consent was obtained from each participant. Students under 18 years of age required informed consent from their parents/guardians to participate in the survey. All procedures followed the guidelines of the Declaration of

Helsinki and were approved by the Ethics Review Committee for Human Science and Technology of Shanghai Jiao Tong University.

The study protocol was approved by the Ethics Review Committee for Human Science and Technology of Shanghai Jiao Tong University, No. H2022248I.

## MEASURES

### Short Grit Scale

Grit was assessed using the eight-item Short Grit Scale (Grit-S) (*Duckworth & Quinn, 2009*). This scale comprised two subscales that measure perseverance of effort (*e.g.*, "I am diligent") and consistency of interest (*e.g.*, "New ideas and projects sometimes distract me from previous ones"), with each subscale consisting of four items. Participants rated their responses on a five-point Likert scale, ranging from 1 (not at all like me) to 5 (very much like me). The total score for grit was obtained by averaging the scores across the eight items, while sub-dimension scores were obtained by averaging item scores within each respective subscale. Higher scores indicate higher levels of grit. Wang's study demonstrated that Grit-S has been validated as effective and reliable among Chinese university students. The fit indices for the two-factor model are relatively high ($GFI = 0.94$, $CFI = 0.87$, $RMSEA = 0.10$), with a Cronbach's alpha coefficient of 0.69 for the total scale, and Cronbach's alpha coefficients of 0.66 and 0.58 for the subscales (*Wang, 2016*). In this study, the Cronbach's alpha coefficient for the overall scale was 0.743. The subscales of consistency of interest and perseverance of effort also showed satisfactory reliability, with Cronbach's alpha coefficients of 0.726 and 0.746, respectively.

### General self-efficacy scale

The General Self-Efficacy Scale (GSES) is a widely used self-report measure of an individual's perception of their own abilities. It was proposed in Germany by Schwarzer and Jerusalem (*Johnston, Wright & Weinman, 1995*). The Chinese version of the GSES was developed by *Zhang & Schwarzer (1995)* and later revised by *Wang, Hu & Liu (2001)*. The scale comprises 10 items (*e.g.*, "I can always manage to solve difficult problems if I try hard enough") and utilizes a four-point Likert scale, with responses ranging from 1 (not at all true) to 4 (completely true). The total self-efficacy score was obtained by averaging the 10 items. Higher scores indicate higher levels of self-efficacy. Wang's study indicated that GSES was applicable to Chinese university students, with a Cronbach's alpha of 0.87. The test-retest reliability was 0.83, and the split-half reliability was 0.82 (*Wang, Hu & Liu, 2001*). In this study, the Cronbach's alpha coefficient for the GSES was 0.877.

### International physical activity questionnaire

The International Physical Activity Questionnaire short form (IPAQ-S) was used to assess individual PA levels. The IPAQ is a widely used and internationally validated tool that has been tested for reliability and validity in 12 countries (*Craig et al., 2003*). The IPAQ-S comprises seven questions that inquire about the time and days spent engaging in different levels of PA per week. The questions were categorized according to their intensity levels, which included vigorous, moderate, and walking. The Chinese version of IPAQ-S is valid

and reliable for assessing PA among college students, with a test-retest reliability of 0.779 and a criterion validity of 0.718 (*Qu & Li, 2004*).

Metabolic Equivalent (MET) values were assigned to three different levels of PA, including 8.0 (MET), 4.0 (MET), and 3.3 (MET) corresponding to high-intensity, moderate-intensity and low-intensity PA for walking, respectively, in the calculation of the total PA score. To calculate the total MET for each participant over seven days, we used the following formula: MET min/week = frequency of vigorous-intensity PA (days/week) * time of vigorous-intensity PA (min/day) * 8.0 (MET) + frequency of moderate-intensity PA (days/week) * time of moderate PA (min/day) * 4.0 (MET) + walking frequency (days/week) * walking time (min/day) * 3.3 (MET).

## Data analysis

The data analysis process involved several steps. Firstly, we implemented data screening and processing. Secondly, we conducted statistical analysis using SPSS 26.0 and reported descriptive statistics, such as means, standard deviations, and independent samples t-tests in Table 1. Scale reliability was assessed using Cronbach's alpha reliability coefficient. Pearson's correlation coefficient was used to quantify associations between the main variables. Finally, to assess mediation effects, we employed model 4 of the Process v4.0 macro and conducted tests for multicollinearity using variance inflation factors (VIFs) and tolerance. For examining moderating effects, we employed model 14 of the macro. In this process, we employed percentile bootstrapping, with a bootstrapping sample size of 5,000 and bootstrapped confidence intervals (*CI*) set at 95%. Significance was determined using a threshold of $p < 0.05$ for all two-tailed tests.

# RESULTS

## Preliminary analysis

Table 1 lists the demographic variables of this study, as well as the subjects' scores for self-efficacy, grit, and PA. The results of independent samples t-tests indicate that male students had higher scores in self-efficacy, perseverance of effort, and PA than female students ($p < 0.01$).

Table 2 shows bivariate correlations between the investigated variables. PA exhibited a positive correlation with self-efficacy ($r = 0.157, p < 0.01$), perseverance of effort ($r = 0.231, p < 0.01$), and consistency of interest ($r = 0.118, p < 0.01$). Furthermore, perseverance of effort and consistency of interest were positively associated with self-efficacy, with correlation coefficients of 0.328 ($p < 0.01$) and 0.316 ($p < 0.01$), respectively. Therefore, this result lays the foundation for the examination of Hypotheses 1–3.

## Testing the mediating role of grit

The Variance Inflation Factor (VIF) and tolerance tests yielded VIF values ranging from 1.05 to 1.20 and tolerance values from 0.83 to 0.86. As VIF < 10 and tolerance > 0.1 denote the absence of multicollinearity (*O'brien, 2007*), the results suggest our data is free from multicollinearity.

Tables 3 and 4 and Fig. 1 show direct and indirect effects of PA on self-efficacy. Parallel multiple mediation analysis found an overall effect of PA on self-efficacy ($\beta = 0.16$,

**Table 1  Descriptive statistics and gender differences in physical activity, grit, and self-efficacy.**

| | Total sample ($N = 3{,}228$) | Male ($n = 2{,}084$) | Female ($n = 1{,}144$) | $p$ |
|---|---|---|---|---|
| Age in years, $M$ ($SD$) | 19.62 (0.84) | | | |
| Male, n (%) | 2084 (64.6) | | | |
| Self-efficacy, $M$ ($SD$) | 2.57 (0.51) | 2.59 (0.52) | 2.53 (0.50) | 0.005 |
| Grit, $M$ ($SD$) | 3.27 (0.62) | 3.28 (0.61) | 3.24 (0.64) | 0.057 |
| Perseverance of effort | 3.62 (0.74) | 3.65 (0.73) | 3.57 (0.74) | 0.005 |
| Consistency of interest | 2.92 (0.81) | 2.92 (0.81) | 2.91 (0.81) | 0.731 |
| Physical activity (METS) | 2427.83 (853.46) | 2479.38 (841.28) | 2333.92 (867.73) | <0.001 |

**Table 2  Pearson correlation analysis for study outcomes.**

| | 1 | 2 | 3 | 4 | 5 | 6 |
|---|---|---|---|---|---|---|
| 1. Age | 1 | | | | | |
| 2. Gender | −0.100[**] | 1 | | | | |
| 3. Self-efficacy | 0.005 | −0.049[**] | 1 | | | |
| 4. Perseverance of effort | −0.074[**] | −0.050[**] | 0.328[**] | 1 | | |
| 5. Consistency of interest | −0.028 | −0.006 | 0.316[**] | 0.293[**] | 1 | |
| 6. Physical activity | −0.037[*] | −0.082[**] | 0.157[**] | 0.231[**] | 0.118[**] | 1 |

Notes.
[*]$p < 0.05$.
[**]$p < 0.01$.

**Table 3  Direct and indirect effects of physical activity on self-efficacy.**

| Outcomes | Predictors | $R$ | $R^2$ | $F$ | $\beta$ | $SE$ | $t$ | $p$ |
|---|---|---|---|---|---|---|---|---|
| Perseverance of effort | | 0.23 | 0.05 | 182.06 | | | | |
| | Physical activity | | | | 0.23 | 0.02 | 13.49 | <0.001 |
| Consistency of interest | | 0.12 | 0.01 | 45.55 | | | | |
| | Physical activity | | | | 0.12 | 0.02 | 6.75 | <0.001 |
| Self-efficacy | | 0.41 | 0.17 | 213.52 | | | | |
| | Physical activity | | | | 0.07 | 0.02 | 4.45 | <0.001 |
| | Perseverance of effort | | | | 0.24 | 0.02 | 14.06 | <0.001 |
| | Consistency of interest | | | | 0.24 | 0.02 | 14.05 | <0.001 |

$p < 0.001$, 95% CI [0.12–0.19]). PA had a direct effect on self-efficacy ($\beta = 0.07$, $p < 0.001$, 95% CI [0.04–0.11]), accounting for 46.73% of the total effect. Perseverance of effort (a1b1) and consistency of interest (a2b2) were mediators, accounting for 35.54% and 17.74% of the total effect, respectively. The total indirect effect of PA on self-efficacy was 0.08 (95% CI [0.07–0.10]), supporting Hypothesis 1 and 2.

**Table 4  Decomposition of total effect, direct effect, and mediating effect.**

| | Path | Effect | SE | 95% CI | | Relative effect |
|---|---|---|---|---|---|---|
| | | | | LL | UL | |
| Total effect | | 0.16 | 0.02 | 0.12 | 0.19 | |
| Direct effect | PA →SE | 0.07 | 0.02 | 0.04 | 0.11 | 46.73% |
| Total indirect effect | PA →Grit →SE | 0.08 | 0.01 | 0.07 | 0.10 | 53.27% |
| *Perseverance of effort (a1b1))* | PA →PE →SE | 0.06 | 0.01 | 0.04 | 0.07 | 35.54% |
| *Consistency of interest (a2b2)* | PA →CI →SE | 0.03 | 0.00 | 0.02 | 0.04 | 17.73% |

**Notes.**

PA, physical activity; SE, self-efficacy; PE, perseverance of effort; Cl, consistency of interest.

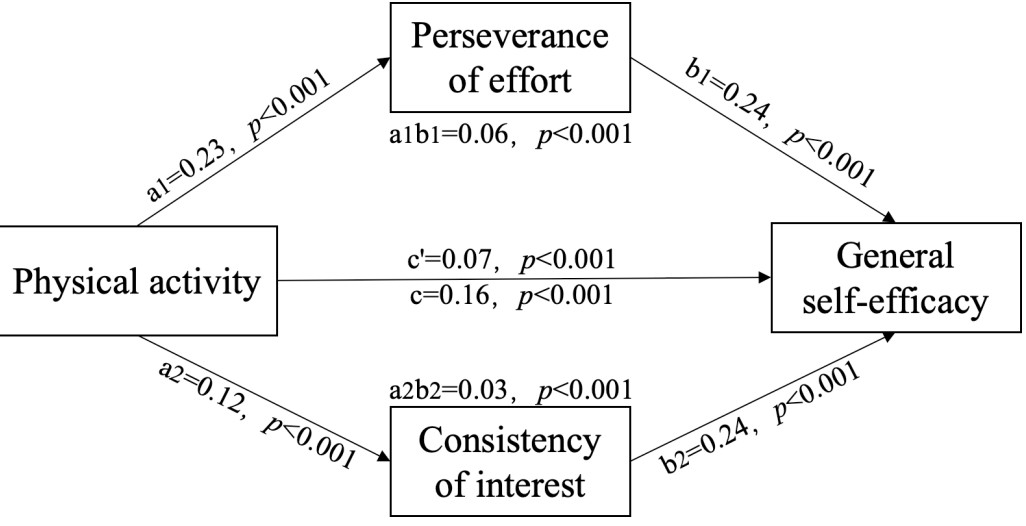

**Figure 1  Direct and indirect effects of PA on self-efficacy.**

### Testing the moderating role of gender

Table 5 shows the results for the moderating role of gender in the second half of the pathway for the mediating effect of grit. The interaction between perseverance of effort and gender predicted self-efficacy ($\beta = 0.08$, $t = 3.27$, $p < 0.01$), indicating that gender plays a moderating role in predicting perseverance of effort on self-efficacy. These findings confirmed Hypothesis 3.

A simple slope analysis explored the interaction effect of perseverance of effort and gender (Fig. 2). Our results revealed that perseverance of effort had a positive effect on self-efficacy for both male students ($\beta = 0.32$, $t = 10.96$, $p < 0.01$) and female students ($\beta = 0.20$, $t = 9.56$, $p < 0.01$). However, the effect was relatively smaller for female students. Further analysis of the conditional mediated effect of PA on self-efficacy indicated that the indirect effect of PA on self-efficacy through perseverance of effort differs between genders (95% CI [0.01–0.09]), as shown in Table 6.

**Table 5  Mediating model with moderating effect.**

| Outcome | Predictors | Fitting indicator | | | Coefficient | | | |
|---|---|---|---|---|---|---|---|---|
| | | $R$ | $R^2$ | $F$ | $B$ | $SE$ | $t$ | $p$ |
| SE | | 0.41 | 0.17 | 109.50 | | | | |
| | Gender | | | | 0.23 | 0.09 | 2.47 | 0.01 |
| | PE | | | | 0.06 | 0.04 | 1.69 | 0.09 |
| | PE × gender | | | | 0.08 | 0.02 | 3.27 | 0.00 |

**Notes.**
SE, self-efficacy; PE, perseverance of effort.

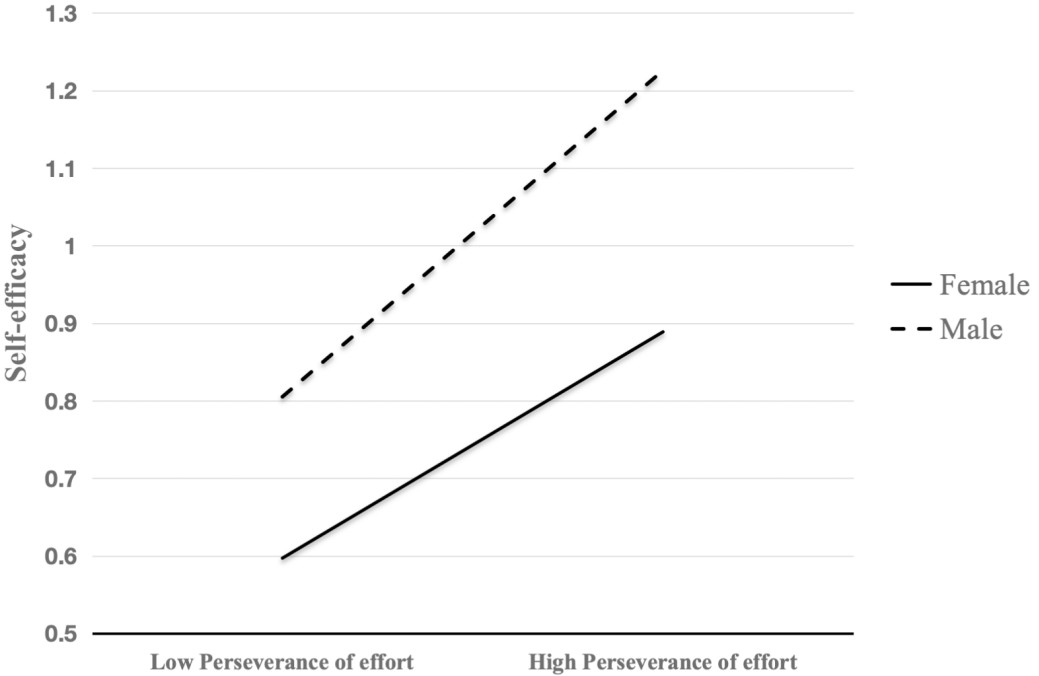

**Figure 2  The interaction effect of perseverance of effort and gender.** Perseverance of effort had a positive effect on self-efficacy for both male students and female students.

**Table 6  Mediating effect of perseverance of effort under different gender.**

| Mediator | Gender | Effect | SE | 95% CI | |
|---|---|---|---|---|---|
| | | | | LL | UL |
| Perseverance of effort | Female | 0.046 | 0.007 | 0.033 | 0.061 |
| | Male | 0.073 | 0.010 | 0.056 | 0.093 |
| | Difference | 0.027 | 0.010 | 0.008 | 0.093 |

## DISCUSSION

This study aimed to examine the mediating role of grit, which includes perseverance of effort and consistency of interest, in the link between PA and self-efficacy among college

students. The findings revealed that PA predicted self-efficacy both directly and indirectly through grit. It was concluded that grit serves as an important role to comprehend the mechanism of PA on self-efficacy, with perseverance of effort acting as a more potent predictor than consistency of interest. Moreover, the impact of perseverance of effort on self-efficacy was observed to be more pronounced in males compared to females.

The present study tested research Hypothesis 1 that PA can directly enhance self-efficacy, which is consistent with the findings of *Miller, Ogletree & Welshimer (2002)* and *Cataldo et al. (2013)*. As the independent variable, PA exhibited a notable role in fostering self-efficacy among college students. Beyond enhancing physical fitness and health status, PA also strengthens students' confidence and self-efficacy through physical exercise and overcoming challenges. This dual effect potentially contributes to their holistic development across academic, social, and personal domains (*Visier-Alfonso et al., 2022*; *Joseph et al., 2014*). Our finding not only underscores the positive psychological impact of PA on individuals but also emphasizes its vital role in the developmental process of university students. Consequently, this study enriches the theoretical underpinnings regarding the enhancement of college students' psychological well-being through PA and highlights the importance for educators to recognize the role of PA in shaping students' positive psychological qualities, ultimately providing robust support for students' overall growth.

This study confirms Hypothesis 2, which suggests that grit plays a mediating role between PA and self-efficacy. This underscores PA's ability to enhance individual grit, thus facilitating the elevation of self-efficacy. Notably, within the domain of grit, perseverance of effort (35.5%) exhibits a greater mediating effect compared to consistency of interest (17.7%). This finding aligns with previous research by *Duckworth et al. (2011)* and *Ciaccio (2019)*, which similarly observed this trend. *Duckworth et al. (2011)* emphasizes the superior predictive power of perseverance of effort over consistency of interest in forecasting academic success among adolescents. Similarly, *Ciaccio's (2019)* investigation reveals a positive correlation between perseverance of effort and exercise self-efficacy, whereas consistency of interest does not exhibit a similar association. This further confirms that PA primarily enhances self-efficacy among university students by fostering their perseverance of effort.

The cultivation of grit is a gradual process, requiring individuals to maintain prolonged effort and resilience in the face of challenges and setbacks, with PA providing an ideal platform. As a proactive lifestyle choice, PA, particularly exercise or sport training, enables individuals to develop grit by continually overcoming obstacles and embracing challenges (*Nothnagle & Knoester, 2022*). This cultivation of grit significantly contributes to the enhancement of self-efficacy. When individuals possess sufficient grit in tackling various tasks and challenges, it instills in them the belief in their capability to accomplish goals. This boost in self-assurance is a manifestation of strengthened self-efficacy. Hence, grit serves as a bridge between PA and self-efficacy, constituting a key variable in the positive influence of PA on self-efficacy.

This study not only enriches our understanding of the interplay among PA, grit, and self-efficacy but also offers a novel perspective and direction for enhancing self-efficacy

among college students. Educators can leverage this insight by designing challenging PA programs to stimulate students' intrinsic motivation, allowing them to practice and enhance grit in real-world scenarios, thereby fostering the elevation of self-efficacy and holistic development.

Neuroscience research further supports our finding that grit acts as a mediator between PA and self-efficacy. Specifically, the dmPFC has been identified as the neural substrate for grit (*Wang et al., 2017*; *Tanji, Shima & Mushiake, 2007*). Dysfunctional PFC can hinder goal-setting and adherence to long-term goals (*Szczepanski & Knight, 2014*), which constitutes the essence of grit. In addition, numerous studies have confirmed that PA is effective in improving the brain's cognitive function. For example, *Kramer et al. (1999)* reported that cognitive switching and compatibility improved significantly in older adults after six months of moderate-intensity aerobic exercise. Similarly, using functional MRI, *Chaddock et al. (2012)* observed that higher fitness levels in 9-10-year-old children were associated with less neural energy use during the Flanker cognitive control study. *Hillman, Erickson & Kramer (2008)* found that adolescents with high aerobic fitness had shorter reaction times for the P3 component of the event-related potential (ERP), leading to better cognitive task performance. Given that cognitive function is a key activity of PFC, grit links the relationship between PA and self-efficacy through the function of the PFC. These neurophysiological study support the enhancement of grit and self-efficacy through PA.

Furthermore, this study discovered the moderating influence of gender in the pathway of perseverance of effort on self-efficacy, thereby confirming the validity of Hypothesis 3. Specifically, the indirect effect of PA on self-efficacy through perseverance of effort was more pronounced in males than in females. Our findings are consistent with *Oriol et al. (2017)*, who found that among primary and secondary students, boys' perseverance of effort was a stronger predictor of self-efficacy. However, it is worth noting that no gender moderation effect was observed in the consistency of interest. A dissertation study on Chinese secondary students yielded results consistent with our independent sample $t$-test, indicating no significant difference in students' consistency of interest but differences in persistence of effort (*Xu, 2022*). This absence of moderation in the consistency of interest pathway may be attributed to similar patterns observed across genders. Gender role socialization theory suggests that personal development may be gendered due to the different roles that males and females take on in the socialization process (*Stockard, 2006*). The Chinese traditional belief that males should be resilient and not give up, while females should be gentle and sensitive, may contribute to the stronger persistence of effort shown by males compared to females (*Song et al., 2021*). In this study, male students had significantly higher persistence of effort than female students. Thus, grasping the impact of gender can assist in pinpointing the mechanistic variations in the ways PA fosters self-efficacy. Similarly, when devising specific physical activity intervention strategies, it's crucial to closely consider the effectiveness of exercise in improving mental health among female college students.

## LIMITATIONS

This study has several limitations. Firstly, its cross-sectional design precludes the establishment of a causal relationship between variables. Secondly, the study's sample was limited to students from one college in Shanghai, which may limit the generalizability and representativeness of the results. Future studies should, therefore, expand the sample to include diverse groups and types of students. Thirdly, the study focused on the intra-individual perspective of the relationships between physical activity, self-efficacy, and grit among college students, However, it overlooks the potential impact of other factors such as interpersonal relationships, school policies, and the local environment on PA and self-efficacy. To comprehensively examine their contributions to self-efficacy, future research should consider these influences.

## CONCLUSIONS

The results of this study reveal that a dual mediation model, incorporating the two dimensions of grit, namely perseverance of effort and consistency of interest, provides an understanding of how physical activity enhances self-efficacy. Furthermore, the study highlights the gender differences in the role of perseverance of effort on self-efficacy, with a stronger effect observed in males compared to females. These findings have important implications for educators to design customized PA interventions that promote grit and self-efficacy in college students.

## ACKNOWLEDGEMENTS

The authors sincerely thank all the participants.

### Funding
This work was supported by the Shanghai JiaoTong University Teaching Development Fund Project (No. CTLD23J0004). The funders had no role in study design, data collection and analysis, decision to publish, or preparation of the manuscript.

### Grant Disclosures
The following grant information was disclosed by the authors:
The Shanghai JiaoTong University Teaching Development Fund Project: No. CTLD23J0004.

### Competing Interests
The authors declare there are no competing interests.

### Author Contributions
- Hongyan Yu conceived and designed the experiments, performed the experiments, analyzed the data, prepared figures and/or tables, authored or reviewed drafts of the article, and approved the final draft.

- Tingfei Zhu conceived and designed the experiments, analyzed the data, prepared figures and/or tables, authored or reviewed drafts of the article, and approved the final draft.
- Jianing Tian performed the experiments, analyzed the data, prepared figures and/or tables, and approved the final draft.
- Gang Zhang performed the experiments, analyzed the data, prepared figures and/or tables, and approved the final draft.
- Peng Wang performed the experiments, analyzed the data, prepared figures and/or tables, and approved the final draft.
- Junxiong Chen analyzed the data, prepared figures and/or tables, and approved the final draft.
- Liqun Shen conceived and designed the experiments, authored or reviewed drafts of the article, and approved the final draft.

### Human Ethics

The following information was supplied relating to ethical approvals (i.e., approving body and any reference numbers):

The University of Shanghai Jiaotong University Leicester granted Ethical approval to carry out the study within its facilities (Ethical Application Ref:H2022248I).

### Data Availability

The raw measurements are available in the Supplementary File.

### Supplemental Information

Supplemental information for this article can be found online at http://dx.doi.org/10.7717/peerj.17422#supplemental-information.

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
