# Peer review of "Physical activity and self-efficacy in college students: the mediating role of grit and the moderating role of gender"

_PeerJ, doi:10.7717/peerj.17422_

## Round 0.1 · original submission · Major Revisions

Thank you for your submission. The reviewers have identified a number of concerns that must be addressed. In particular the Introduction as well as the statistical analysis.

Reviewer 1 ·

Basic reporting

The authors highlight a significant topic: students' sense of self-efficacy, physical activity, the mediating role of grit, and the moderating role of gender. The revieved paper focuses on the relationship between physical activity and self-efficacy among college students in China, as well as on the mediator role of grit and the moderating role of gender in shaping this relationship. However, i have some concerns about the thereotical background and other methodological issues. Here are some of them:
1. The content of the introductory part of the paper should be better organized. The way the authors combine theoretical background with the study's assumptions makes it hard for the reader to follow. It should be better to explain concepts in a hierarchy from general to specific. For example, any reader could better understand the main reason for the authors' choice of topics if they organised the relationships between physical activity and self-efficacy. Therefore, the introduction should be better organized regarding the mediating and moderating effect of grit and gender. There should also be gender-related empirical studies and theoretical background.
2. Empirical research on the subject in the Chinese sample should be reviewed and critically analyzed. The originality of the research should be emphasized by focusing more on what the authors examine differently from previous studies. The introduction to the study is very short. Explanations on why intermediary and regulatory influence are addressed are inadequate.
3. I think there is no need for hypothesis 1 about the relationship between variables. It would be more appropriate to write hypotheses focusing on mediation, moderation, direct and indirect effects.

Experimental design

4. The statistical evaluation of the significance of the parameters proposed by the Author can't be validly used with non-random samples as they may lead to erroneous results and conclusions. Statistical inference is valid only if probabilistic sampling is undertaken. Non probabilistic samples cannot be used to make reliable statistical inferences and it is pointless to interpret the p-value of such samples because it could mislead the reader. Therefore, verifying the hypothesis on the basis of the calculations presented is, in my view, highly questionable.
5. I suggest to the author to revise the empirical part of the study and to avoid strong statements about (statistically) significant (or not) relationships between the analysed variables in the discussion of the results.
6. Another concern is the sampling method. Convenient sampling method is highly questionable to make reliable inferences. The authors did not report the number of students in the population, the rate of participation in the scales, the results of the power analysis on how many people they needed to reach.
7. In line 124, the authors indicated; "We eliminated invalid responses, including incomplete responses, inconsistencies, unreliability, and data points outside the defined range of acceptable values." However the data elimination type is unclear. How did you eliminate the invalid responses, how did you decide them? How did you clean the data? Deleted them casewise or listwise?

Validity of the findings

8. In preliminary analysis it should be done the dfa of the scales among current sample.

·

Basic reporting

Overall, the language and the structure of the article are good. The manuscript is written in English and uses clear, unambiguous, technically correct text. It conforms to professional standards of courtesy and expression. The structure of the article conforms to an acceptable format of standard sections. Figures and tables are relevant to the content of the article and are appropriately described and labeled. All appropriate raw data has been made available. The submission is ‘self-contained’ and includes all results relevant to the hypothesis authors established.
However, the introduction needs revising. The article does not include sufficient introduction and background to demonstrate how the work fits into the broader field of knowledge. Relevant prior literature seem cited but previous findings need to be introduced in more detail.
I believe that the introduction does not provides the strong rational to conduct the study. This study proposed and tested the moderated mediation model in which PA would affect self-efficacy via grit, which would be moderated by gender. It is not clear in the manuscript in its current form why such a moderated mediation model is hypothesized. Your introduction needs more detail. I have two comments/suggestions. First, I suggest that you improve the description at lines 88-99. If my reading is correct, what authors want to suggest in this paragraph is that grit may mediate the relationship between PA and self-efficacy. The topic (first) sentence is “Grit is a critical factor associated with PA and self-efficacy, which plays a vital role in determining student success”. This sentence does not directly convey what authors want to suggest. I would suggest rewriting. The description at lines 90-92 (ie, Additionally, grit is positively associated with domestic and leisure-time physical activity … [28]) seems confusing. This is because this sentence implies that grit increases PA, which might be an opposite direction to the hypothesized direction. I would suggest deleting this sentence.
Second, I suggest that you improve the description at lines 101-106. This paragraph refers to potential effects of gender on the relations among PA, grit, and self-efficacy. Gender differences are not clearly noted here. It is necessary to describe previous findings on gender effects. These additions will make the research questions and hypotheses clearer.

Experimental design

Overall, the experimental design is good. The submission reports original primary research and is within Aims and Scope of the journal. It defines the research question, which is relevant and meaningful. The knowledge gap being investigated is identified. The investigation has been conducted rigorously and to a high technical standard. The research has been IRB-approved.
However, the introduction can be improved. For making research questions clearer, it is necessary to put information about previous findings related to the study in detail and to improve the flow of some paragraphs. It is not clear how the study contributes to filling that gap. Statements need to be made in Introduction and Discussion.

Validity of the findings

The data on which the conclusions are based is provided. The data is robust, statistically sound, and controlled. The conclusions are certainly connected to the original question investigated. However, the conclusion is too strong and needs to be revised. The sentence, “The results of this study reveal that a dual mediation model, incorporating the two dimensions of grit, namely perseverance of effort and consistency of interest, provides a comprehensive understanding of how physical activity enhances self-efficacy” is too strong. The current results can enhance understanding of how physical activity enhances self-efficacy, but can not provide a comprehensive understanding. This is because there are many possible variables other than grit affecting these pathways.

Additional comments

Thank you very much for giving me the opportunity to review the manuscript entitled “Physical activity and self-efficacy in college students: the mediating role of grit and the moderating role of gender”. I have read it with interest. This study examined the moderated mediation model in which physical activity would enhance general self-efficacy among college students via grit, which would be moderated by gender. It was found that physical activity had an indirect effect through the two dimensions of grit: on self-efficacy. Gender was found to moderate the effect of grit on self-efficacy, with the effect stronger in males. The results add new information about how physical activity and general self-efficacy are related to each other among college students. I feel that there is room for improvement. I have listed my other comments with my hope that the comments will be helpful for authors.

Major comments:
1) The flow of the sentences can be improved
I have some sentences which I believe authors might improve its flow.

Line 62-73. I can’t follow the flow of this paragraph. If my understanding is correct, what authors want to suggest in this paragraph is that the mechanisms underlying the relationship between PA and self-efficacy remain to be studied. If so, the first sentence of this paragraph needs to be rewritten. This paragraph begins with the sentence “Despite the potential of PA to enhance self-efficacy, both are subject to various activities and situations”. I am not sure how this sentence is related to the lack of knowledge about the mechanisms underlying the relationship between PA and self-efficacy.

Line 225-231. I can’t follow the flow of this paragraph. If my understanding is correct, authors mention that perseverance of effort is a stronger predictor than consistency of interest. They concluded that previous results had provided further support for the hypotheses presented in this study. I am not sure which hypotheses authors are referring to. Also, I am confused because three hypotheses are not referring to strength of predictive power of each aspect of grit. Did you have any hypotheses about differences in strength of association with self-efficacy between two aspects of grit? If so, I recommend authors to describe it.

Line 232-242. I can not understand what authors want to suggest in this paragraph. Most space is used for introducing previous findings, but it is needed to document why the current findings are related to the suggestion that the crucial role of perseverance in PA merits deeper investigation.

2) The implications of the current findings are not described sufficiently.
Overall, the sentences in the Discussion part are used for describing previous findings related to the study. However, in this part, it is needed to describe what the implications of the current results are. Citating previous findings and comparing to the current results are one of ways to discuss the implications.

Minor comments:
1) Line 135, Grit Scale. It is necessary to add information about reliability and validity of the Grit Scale.
2) Line 146, General Self-Efficacy Scale. Ditto General Self-Efficacy Scale.
3) Appendix, strobe checklist, “Page No”. If appropriate, “Line No” might be better.

---

## Round 0.2 · Minor Revisions

Your article needs some final minor revisions.

• Only use acronyms if you use it at least three times in the article.
• Please include how parental consent was obtained for students aged 17 years old.
• Statistical symbols should be in italics.
• Line 149: remove the word “ancient”.
• Lines 151-152: Please add citation and page number for the quotation. "perseverance and passion in the face of obstacles and adversity in the pursuit of long-term goals,".
• Line 158: Name the scholar.
• Line 173: Remove the word “Korean”
• Line 311: Change GESE to GSES.
• Line 405: Spelling should be “study”.
• Lines 438-442: It is not necessary to discuss the mice study.

Reviewer 1 ·

Basic reporting

No comment

Experimental design

No comment

Validity of the findings

No comment

Additional comments

No comment

·

Basic reporting

Overall, the language and the structure of the article are good. The manuscript is written in English and uses clear, unambiguous, technically correct text. It conforms to professional standards of courtesy and expression. The structure of the article conforms to an acceptable format of standard sections. Figures and tables are relevant to the content of the article and are appropriately described and labeled. All appropriate raw data has been made available. The submission is ‘self-contained’ and includes all results relevant to the hypothesis authors established. I asked authors to revise the introduction. The previous version of the article did not include sufficient introduction and background to demonstrate how the work fits into the broader field of knowledge. Relevant prior literature was cited but previous findings need to be introduced in more detail. The revised version of the manuscript does include sufficient introduction and background and previous findings are explained in detail.

Experimental design

Overall, the experimental design is good. The submission reports original primary research and is within Aims and Scope of the journal. It defines the research question, which is relevant and meaningful. The knowledge gap being investigated is identified. The investigation has been conducted rigorously and to a high technical standard. The research has been IRB-approved.

Validity of the findings

The data on which the conclusions are based is provided. The data is robust, statistically sound, and controlled. The conclusions are well stated, certainly connected to the original question investigated, and limited to supporting results.

Additional comments

Thank you very much for giving me the opportunity to review the revied manuscript. I have carefully reviewed the revised manuscript and have found that all my concerns have been addressed.

---

## Round 0.3 · accepted · Accept

Thank you for your revised submission. I am satisfied that you addressed the remaining concerns of the reviewer, and am happy to accept your paper for publication